# MALDI-TOF MS Limits for the Identification of Mediterranean Sandflies of the Subgenus *Larroussius*, with a Special Focus on the *Phlebotomus perniciosus* Complex

**DOI:** 10.3390/microorganisms10112135

**Published:** 2022-10-28

**Authors:** Antoine Huguenin, Bernard Pesson, Matthieu L. Kaltenbach, Adama Zan Diarra, Philippe Parola, Jérôme Depaquit, Fano José Randrianambinintsoa

**Affiliations:** 1EA 7510 ESCAPE, USC VECPAR, ANSES, SFR Cap Santé, Université de Reims Champagne-Ardenne, 51096 Reims, France; 2Laboratoire de Parasitologie, Pôle de Biologie Territoriale, CHU, 51100 Reims, France; 3IRD, AP-HM, SSA, VITROME, IHU-Méditerranée Infection, Aix Marseille University, 19-21 Boulevard Jean Moulin, 13005 Marseille, France

**Keywords:** *Phlebotomus*, *Larroussius*, species complex, MALDI-TOF MS, identification, high-throughput identification, molecular biology

## Abstract

*Leishmania infantum* is the agent of visceral leishmaniasis in the Mediterranean basin. It is transmitted by sandflies of the subgenus *Larroussius*. Although *Phlebotomus perniciosus* is the most important vector in this area, an atypical *Ph. perniciosus* easily confused with *Ph. longicuspis* has been observed in North Africa. MALDI-TOF MS, an important tool for vector identification, has recently been applied for the identification of sandflies. Spectral databases presented in the literature, however, include only a limited number of *Larroussius* species. Our objective was to create an in-house database to identify Mediterranean sandflies and to evaluate the ability of MALDI-TOF MS to discriminate close species or atypical forms within the *Larroussius* subgenus. Field-caught specimens (*n* = 94) were identified morphologically as typical *Ph. perniciosus* (PN; *n* = 55), atypical *Ph. perniciosus* (PNA; *n* = 9), *Ph. longicuspis* (*n* = 9), *Ph. ariasi* (*n* = 9), *Ph. mascittii* (*n* = 3), *Ph. neglectus* (*n* = 5), *Ph. perfiliewi* (*n* = 1), *Ph. similis* (*n* = 9) and *Ph.* *papatasi* (*n* = 2). Identifications were confirmed by sequencing of the mtDNA CytB region and sixteen specimens were included in the in-house database. Blind assessment on 73 specimens (representing 1073 good quality spectra) showed a good agreement (98.5%) between MALDI-TOF MS and molecular identification. Discrepancies concerned confusions between *Ph. perfiliewi* and *Ph. perniciosus*. Hierarchical clustering did not allow classification of PN and PNA. The use of machine learning, however, allowed discernment between PN and PNA and between the lcus and lcx haplotypes of *Ph. longicuspis* (accuracy: 0.8938 with partial-least-square regression and random forest models). MALDI-TOF MS is a promising tool for the rapid and accurate identification of field-caught sandflies. The use of machine learning could allow to discriminate similar species.

## 1. Introduction

*Leishmania infantum* Nicolle is the main agent of visceral (VL) and cutaneous (CL) leishmaniasis. In the Mediterranean basin, the Middle East and East Africa, this zoonosis is transmitted by the bite of an infected female Phlebotomine sandfly belonging mainly to the genus *Phlebotomus* Rondani & Berté, subgenus *Larroussius* Nitzulescu [1]. In the case of leishmaniasis due to *L. infantum*, domestic dogs and wild foxes play the role of vertebrate hosts. Although rabbits could play the role of alternative reservoir hosts locally [2], the role of cats as alternative vertebrate hosts has been mostly neglected to this date [3].

In the Western part of the Mediterranean basin, *L. infantum* proven vectors are *Ph. perniciosus* Newtsead, *Ph. longicuspis* Nitzulescu, *Ph. langeroni* Nitzulescu, *Ph. perfiliewi* Parrot and *Ph. ariasi* Tonnoir, all belonging to the subgenus *Larroussius*. The most widespread and abundant species in this area and major vector is *Ph. perniciosus*. *Ph. perfiliewi* is also widespread, but with a lower frequency. The distribution areas of *Ph. longicuspis*, *Ph. langeroni* and *Ph. ariasi* are limited, but they can also act as vectors locally [4]. 

The *Ph. perniciosus* complex Nitzulescu [5] includes two species: *Ph. perniciosus* and *Ph. longicuspis*. First identified in Malta, *Ph. perniciosus* is widely distributed throughout the western Mediterranean Basin: from Croatia to Portugal in Europe, and from Libya to Morocco in North Africa where it is predominantly found in sub-humid and semi-arid bioclimate zones [6]. *Phlebotomus longicuspis* distribution is restricted to North Africa where it is most frequently recorded in semi-arid, arid, and per-arid bioclimate zones (Rioux et al., 1984). Its identification in southern Spain [7,8] remains doubtful [9]. Indeed, females have long been considered undistinguishable. Their identification, based on the morphology of the base of the basal dilatation of their spermathecal ducts [10], is only possible for well-trained entomologists. Identification of the males is based on the top of the parameral sheath, which is bifurcated in *Ph. perniciosus,* whereas it is simple, lightly curved and pointed in *Ph. longicuspis*. Moreover, *Ph. perniciosus* male specimens exhibit ten to 16 internal setae on their gonocoxite, whereas *Ph. longicuspis* exhibits 15 to 29 gonocoxal setae. 

An atypical form of *Ph. perniciosus* (PNA as compared to typical PN *Ph. perniciosus*), with a simple and curved distal portion of parameral sheath instead of bifurcated, which can be confused with *Ph. longicuspis,* has been described in the western part of Mediterranean region [9]. Males of the two species can, however, be morphologically differentiated by the number of coxite setae, as explained above [11]. No morphological character has been described to date to identify PNA females from PN females. Isoenzymatic analysis classified PNA as *Ph. perniciosus*, and sequence analysis of the COI fragment of mtDNA have shown a distinct haplotype for PNA [9,12]. However, the mtDNA PNA haplotype was also observed for two males with PN morphology, raising concerns concerning its reliability to correctly identify PNA [9]. The PNA form has been described in Morocco [9,13,14,15,16,17], Algeria [11] and Tunisia [12,18,19]. Probable atypical *Ph. perniciosus* forms, intermediate between *Ph. perniciosus* and *Ph. longicuspis,* have also been described in Spain under the *Ph. longicuspis* name [18,19]. The PNA form reported in Morocco seems to correlate positively with altitude and humid and sub-humid climates [15,16]. Zarrouk et al. [16] reported that the geographical repartition of PNA is negatively correlated with that of human VL, whereas the correlation was positive for PN. In Tunisia, Ghrab et al. [20] described the predominance of PNA in South-Eastern Tunisia where no *L. infantum* VL has been reported and its absence in foci of the disease. These observations raise the question as to the role of PNA in the transmission of VL due to *L. infantum*. 

MALDI-TOF MS is currently an important tool for the identification of arthropods of medical or veterinary importance [21]. It has also been successfully applied to the identification of sandflies [22,23,24,25]. It allows the identification of immature stages [26] and blood meals [27,28] in engorged females. 

Few studies investigated field-caught sandflies belonging to the *Ph. perniciosus* complex. In 2015, Mathis et al. reported [23] on the identification of sandflies by means of MALDI-TOF MS with a database of 20 species obtained from specimens captured in the field or from colonies. A grouping according to the origin (colonies or field) of the spectra of *Ph. perniciosus* was noted. Discrepancies were reported for the *Larroussius* subgenus, particularly for a specimen identified morphologically as *Ph. galilaeus* whose spectrum matched with that of *Ph. perniciosus*. However, specimens of *Ph. perniciosus* did not come from areas where PNA are described.

Our objective was to evaluate the performance of MALDI-TOF MS to identify species of the genus *Larroussius* and to see if this technique can distinguish between different morphotypes of *Ph. perniciosus*.

## 2. Materials and Methods

### 2.1. Sandfly Sampling

Sandflies were collected using CDC miniature light traps (John W. Hock company, Gainesville, FL) as explained in Table 1. All specimens were initially stored in liquid nitrogen at −80 °C since 2015. They were processed between 2019 and 2022.

Our sample included typical *Ph. perniciosus* from several countries: Portugal, Spain, France, Italy, Malta and Morocco; PNA *Ph. perniciosus* from Morocco, as well as a few other *Larroussius* (i.e., *Ph. perfiliewi* and *Ph. ariasi*). Several other species belonging to the subgenera *Transphlebotomus*, *Phlebotomus* and *Paraphlebotomus* were also added to our sample as out-groups (Table 1). 

Briefly, specimens were dissected on cold table (−20 °C) under a stereomicroscope (Olympus, Japan). The head and genitalia of individual sandflies were cut off and processed for morphological analysis (see below). The body of each specimen was separated into thorax and abdomen, then stored individually in 1.5 mL sterile tubes at −20 °C.

### 2.2. Morphological Analysis

Head and genitalia were processed in individual vials. Soft tissues were lysed in a bath of 10% KOH overnight, washed with distilled water, bleached in Marc-André solution overnight and mounted between a microscope slide and cover slide in Euparal^®^ for species identification after dehydration in successive alcoholic baths. The species were identified by observation of the head and genitalia under a BX50 microscope (Olympus, Japan). 

### 2.3. Molecular Analysis

Genomic DNA was extracted from the abdomen of individual sandflies using the QIAmp DNA Mini Kit (Qiagen, Germany) following the manufacturer’s instructions, except for crushing sandfly tissues with a piston pellet (Treff, Switzerland), and using an elution volume of 150 µL [29].

A fragment of 550 bp of cytochrome b (Cyt b) (using the C3B-PDR: 5′-CAYATTCAACCWGAATGATA-3′ and N1N-PDR: 5′-GGTAYWTTGCCTCGAWTTCGWTATGA-3′ primers) was amplified by PCR [24]. Amplicons were analyzed by electrophoresis in 1.5% agarose gel containing Gel Green at a concentration of 0.005% *v/v*. Direct sequencing in both directions was performed using the primers used for DNA amplification.

Sequences obtained during the present study have been deposited in GenBank under the numbers OP617463 to OP617553 and compared to reference sequences (Pesson et coll., 2004 [9]).

A phylogenetic tree was inferred using the Maximum-Likelihood method with Mega11 [30]. The HKY (Hasegawa-Kishino-Yano) model was selected as the most appropriate model of DNA evolution. Statistical support of internal was evaluated by bootstrapping (100 replicates). 

### 2.4. MALDI-TOF MS Analysis

Thoraxes, including legs of the specimens, were placed in 10 µL of formic acid (Sigma-Aldrich, Lyon, France) and homogenized with a teflon pestle in a microtube. After addition of 10 µL of acetonitrile (Sigma-Aldrich) and centrifugation for 2 min at 10,000 rpm, 1 μL of supernatant was deposited on a 96-well steel target plate. Four spots were deposited for each specimen. After complete drying, 1 μL of HCCA matrix (Bruker Daltonics) saturated solution in organic solvent solution (Sigma-Aldrich) was added. After complete drying at room temperature, plates were then analyzed in a Bruker Microflex LT MALDI–TOF spectrometer (Bruker Daltonics, Champs-Sur-Marne, France). Spectra acquisition was repeated 8 times for each well using the built-in Bruker method “MBT_BTS_Validation_AutoX”. A bacterial test standard was used for instrument calibration. Lastly, spectra were visualized using FlexAnalysis v3.4 and imported in Biotyper Compass Explorer v4.1.100 for analysis.

The Bruker “MALDI Biotyper Preprocessing Standard Method” was used for the creation of main spectra profiles (MSP). High-quality spectra were selected for incrementing the local database (at least 10 good quality spectra by MSP). Hierarchical cluster analysis (HCA) was performed using a correlation method and the Ward algorithm for clustering with the MSP dendrogram tool of Compass Explorer. Sensitivity and specificity were assessed by blind testing the in-house MSP database against spectra from 77 specimens not used for MSP creation. The cut-off value of log-score value (LSV) was determined using receptor-operated-channel curve (ROC curve) analysis with the R pROC package [31]. Our spectra were also tested against the sandflies’ MSP of the Vitrome UMR D 257 (Pr. Philippe Parola).

Machine learning discrimination between PN and PNA was based on custom R scripts based on the caret package [32] using four algorithms: k nearest neighbour (KNN), partial least-square discriminant analysis (PLS), random forest analysis (RF), and extreme gradient boosting machine (xgbTree). Bruker FID spectra files were imported using the MALDIQuantForeign package [33]. Preprocessing and selection of high quality spectra was performed by using the screenSpectra function from the MALDIrppa package [34]. Data were partitioned into training, validation and test datasets. The test dataset contained spectra from specimens that did not belong to the training and validation datasets. Repartition of spectra between different group is reported in Appendix A.

Model training was performed on the training dataset with the KNN, PLS, RF and xgbTree caret [32] methods, with repeated cross validation (ten folds and ten repeats). LogLoss metrics were used to evaluate model performance.

## 3. Results

### 3.1. Entomological Investigation

Atypical Phlebotomine sandflies were clearly identified at the specific level according to their morphological characters. The unequivocal identification of the members of the *Ph. perniciosus complex* was based on the observation of the top of the parameral sheath and the number of gonocoxal internal setae.

### 3.2. Molecular Identification

Morphological identification of all specimens included in our study was confirmed by cytb mtDNA sequencing with homologies >99% versus reference sequences deposited in Genbank. A phylogenic tree inferred from the Maximum-Likelihood method is presented in Figure 1.

### 3.3. MALDI-TOF MS Analysis

#### 3.3.1. MALDI-TOF Analysis and Construction of an In-House Database

Non-flat-line, usable spectra were obtained from 95 out of 99 specimens (95.95%). Visual assessment of acquired spectra from all *Phlebotomus* species using Flex analysis showed consistent and reproducible spectra for each species. Spectra of five representative specimens are presented in Figure 2. MALDI-TOF MS spectra of *Ph. perfiliewi* and *Ph. perniciosus* PN were very similar whereas *Ph. perniciosus* PNA displayed a different pattern of peaks intensities.

MSP was obtained for 90 specimens. Cluster analysis of 64 specimens is presented in Figure 3, showing good separation between *Ph. perniciosus* complex and other species. The spectra of *Ph. perfiliewi* (MT23) was mixed with spectra of *Ph. perniciosus*. For *Ph. perniciosus,* no clear clustering according to morphotype (PNA/PN), sex or geographical origin (Iberic vs. Italian-Maltese sublineage) was identified.

High-quality spectra from 17 specimens were chosen for creating an in-house MSP library: *Ph. perniciosus* (PN: 6 MSP; PNA: 2 spectra), *Ph. longicuspis lcx* (2 MSP), *Ph. longicuspis lcus*, *Ph. ariasi* (*Ph. mascittii*, *Ph. neglectus*, *Ph. papatasi*, *Ph. perfliliewi* (1 MSP for each species).

#### 3.3.2. Blind Test Evaluation

##### In-House Database

3496 spectra acquired from 73 specimens were tested against the upgraded homemade reference database. Determination of LSV threshold was based on molecular identification using ROC curve (Figure 4). The optimal threshold value, as determined by the Youden J statistic, was 1.665, with a sensitivity of 93.3% and a specificity of 82.5%. In order to compare our results with those of other studies [25], we used an LSV cut-off value of 1.7 for identification at the species level instead achieving a specificity of 82.5% and a sensitivity of 93.0%. Using a LSV ≥ 2 for specific identification did not significantly improve specificity (84.57%) but decreased sensitivity (90.34%).

Among the 3496 spectra, 1073 (31%) reached a log score > 1.7, which was considered a probable identification at the species level. Six specimens (8.2%) displayed spectra that did not reach the LSV cut-off. One of these was MT08, a specimen of *Ph similis* not present in the database. The confusion matrix (Table 2) revealed a good agreement (98.5% at spectra level, 97.1% at specimen level) between MALDI-TOF MS and molecular identification.

##### Marseille Database

Spectra acquired from our 73 specimens were matched against the Marseille MSP database. 894 reached the cut-off LSV-value (25.57%). Agreement between molecular biology and MALDI-TOF was 97.87% (Table 3). The 19 discrepant spectra were from *Ph. perniciosus* (18/19) (confused with *Ph. perfiliewi* (*n* = 13)) or *Ph. tobbi* (*n* = 5), and *Ph. longicuspis* (1/19) (confused with *Ph. perfiliewi*). The Marseille database did not allow the discrimination of lcux from lcx haplotypes of *Ph. longicuspis*.

Marseille and Reims (in-house) databases were also matched against spectra from the samples of Mhaidi et al. [36], kindly provided by Petr Halada. Results are presented in Appendix A. In spite of the high quality of the spectra, only 5 spectra out of 18 reached the 1.7 LSV cut-off value with the Reims in-house database and none with the Marseille database. For spectra reaching the cut-off value, two specimens of *Ph. longicuspis* were identified as *Ph. perniciosus* with the Reims database. Interestingly, the second identification was *Ph. longicuspis* in both cases.

#### 3.3.3. PNA Identification Using Machine Learning 

Spectra from *Ph. longicuspis* (lcx and lcus haplotypes), PNA and PN haplotypes of *Ph. perniciosus* were partitioned into (I) a training dataset (500 spectra from 53 specimens), (II) a validation dataset (124 spectra from 30 specimens) and (III) a test dataset (273 spectra from 11 specimens not used for training or validation). The training dataset was balanced concerning the two *Ph. longicupis* haplotypes, PN and PNA to prevent overfitting with 125 spectra for each taxa. Use of 4 common machine learning algorithms allowed to discriminate spectra from the test dataset as *Ph. perniciosus* PNA/PN and *Ph. longicuspis* lcx and lcus with good reliability (Figure 5). The best results were obtained with the RF or PLS algorithms attaining an accuracy of 89.38%. Most of the confusions arose between *Ph. longicuspis lcus* and *lcx* haplotypes, particularly for the K Next-Neighbour (KNN) and eXtreme Gradient Boosting (XGBoost) models.

## 4. Discussion

MALDI-TOF MS allows to discriminate *Larrousius* from non-*Larroussius* subgenus sandflies efficiently. Clustering according to sex, as reported by Lafri et al. [24], was not encountered in our specimens. Within the *Ph. perniciosius* complex, no classification errors were noted with our in-house database for the *Ph. neglectus* and *Ph. ariasi species*. Confusions between *Ph. perfiliewi* and *Ph. perniciosus sensu stricto* were, however, endured. Spectra of these closely related species were visually indistinguishable. This confusion has also been reported by other authors. In 2016, Lafri et al. [24] described the application of MALDI-TOF MS for the identification of sandflies caught in the field. Their spectral database included *Ph. perniciosus* (PN form), *Ph. longicuspis* and *Ph. perfiliewi* species of the subgenus *Larroussius*. They reported very similar observations between MALDI-TOF MS spectra for these two species, and a difficulty in identifying these two species using Bruker MALDI-Biotyper. 

The Marseille database allowed the correct classification of 7 out of 14 *Ph. perfiliewi* spectra. The remaining spectra were also classified as *Ph. perniciosus*. Unfortunately, only one specimen of *Ph. perfiliewi* was collected, thus not allowing the use of machine learning algorithms to discriminate between this two sibling species using MALDI-TOF MS. The inclusion of new specimens of *Ph. perfiliewi* would, however, allow the identification of *Ph. perfiliewi* subspecies [37]. Both databases did not permit reliable identification of spectra obtained from a previous study by Mhaidi et al. [36]. This result illustrates the variability commonly seen between laboratories using MALDI-TOF MS technology. Our sample preparation differed from the one used by our colleagues in Marseille, by using the thorax and legs without wings with grinding in formic acid before adding acetonitrile, instead of using a mixture of these two substances. Mhaidi et al. only used thoraxes with grinding in formic acid alone, followed by mixing with sinapinic acid MALDI matrix. The specimen preparation protocol, although based on the same principles, seems to have a significant impact on the spectra obtained, which limits the portability of existing MSP databases. The protein composition of the wings or legs does probably affect the MALDI-TOF MS spectra, explaining at least partly some of the discrepancies noted between different databases. The spectral variability can also be explained by the use of different spectrometers, although they share the same brand, technology and calibration procedures with identical Bruker bacterial standards. Standardization of protocols thus appears necessary to allow the exchange of spectra between MALDI-TOF MS users. Automatic processing would also be particularly interesting to quickly obtain a large quantity of spectra with a high reproducibility [21].

No clustering according to the haplotype was observed for PN spectra using hierarchical classification, as well as no discrimination between populations of *Ph. perniciosus* as previously emphasized. Pesson et coll. [9] demonstrated that the Italian-Maltese lineage could be discriminated from the Iberian lineage by both isoenzyme analysis and cyt b sequences analysis. The French population of *Ph. perniciosus* is in an intermediate position in relation to the hypothesis that its settlement could be a consequence of a late dispersion following the last glaciation [38]. Moreover, the same authors separated the European lineages from the North African ones when MALDI-TOF MS with hierarchical classification could not achieve this. 

Good discrimination of PN/PNA and *Ph. longicuspis* haplotypes using MALDI-TOF MS has been obtained with machine learning. *Ph. perniciosus* whole genome sequencing could allow recognition of the protein origin of these discriminant peaks and could thus provide information on the genetic status of these two forms. Machine learning is an interesting approach to extract information from MALDI-TOF MS spectra. It has been used for bacteria identification, antibiotic resistance prediction [39,40,41], malaria transmission prediction in *Anopheles* mosquitoes [42] and *Aspergillus* clones identification [43]. It has a great potential to discriminate between MALDI-TOF spectra of insects’ close species. The need of a large sample size with a similar number of specimens between classes, however, is an important limitation. In our study, a training dataset with close to 200 spectra in each class was sufficient to establish models capable of robust prediction. Our study is also limited by the absence of an external validation dataset with spectra acquired from a different MALDI-TOF device in another laboratory. The wide acceptance of the model and the reproducibility of the prediction is thus somewhat limited. Nevertheless, we used a test dataset with specimens that had never been used for training and spectra acquired during a new session held 10 months after the training and validation set. The analytical performance of the different algorithms in the test dataset lead us to think that overfitting was limited in our case. 

It is important to note that the cyt b PNA (pern06) haplotype was also found in two PN males from Morocco and Tunisia [9]. Benabdennbi et al. also noted that a unique female could lay eggs of both PN and PNA males [44]. This implies that the haplotypes described for PNAs are not a completely reliable marker. In this study, we used, for the training dataset, spectra from males identified both morphologically and molecularly as PNA and females of the PNA haplotypes. This approach allowed a reliable identification in the test dataset of morphologically identified males PNA and of females of the same haplotype. MALDI-TOF MS could thus be used for rapid identification of females corresponding to the PNA haplotype. This could be of epidemiological interest since the presence of a PNA population seems associated with the absence of foci of *L. infantum* VL [16,20].

## 5. Conclusions

The collection of a large number of specimens is necessary to improve the discriminatory power of MALDI-TOF and the identifiability of *Phlebotomus* specimens, especially in the case of *Ph. longicuspis* and *Ph. perfiliewi s. l*. Unfortunately, we could not find any other specimen of the latter species in the processed collections. Hierarchical classification of spectra cannot cluster the geographical populations of *Ph. perniciosus* as cyt b sequences. Machine learning, however, allowed reliable discrimination between PN and PNA subspecies, *Ph. longicuspis* lcx and lcus haplotypes. It is a promising approach to distinguish sibling species with similar spectral profiles. The differentiation of these four taxa by morphology requires a trained microscopist and a good dissection of genitalia. The use of chaetotaxy sometimes does not permit differentiation of PNA from *Ph. longicuspis* when the specimen presents 15 or 16 gonocoxal setae. MALDI-TOF is a tool that does not require a formal training in entomology which can be used to quickly screen a collection of frozen sandflies. The use of MALDI-TOF in routine testing for closely related sandflies species identification remains however difficult at the present time. as methods used by laboratories worldwide need some type of standardization. Morphological expertise remains particularly important and, in many cases, identification by molecular biology remains necessary for reliable diagnosis.

## Figures and Tables

**Figure 1 microorganisms-10-02135-f001:**
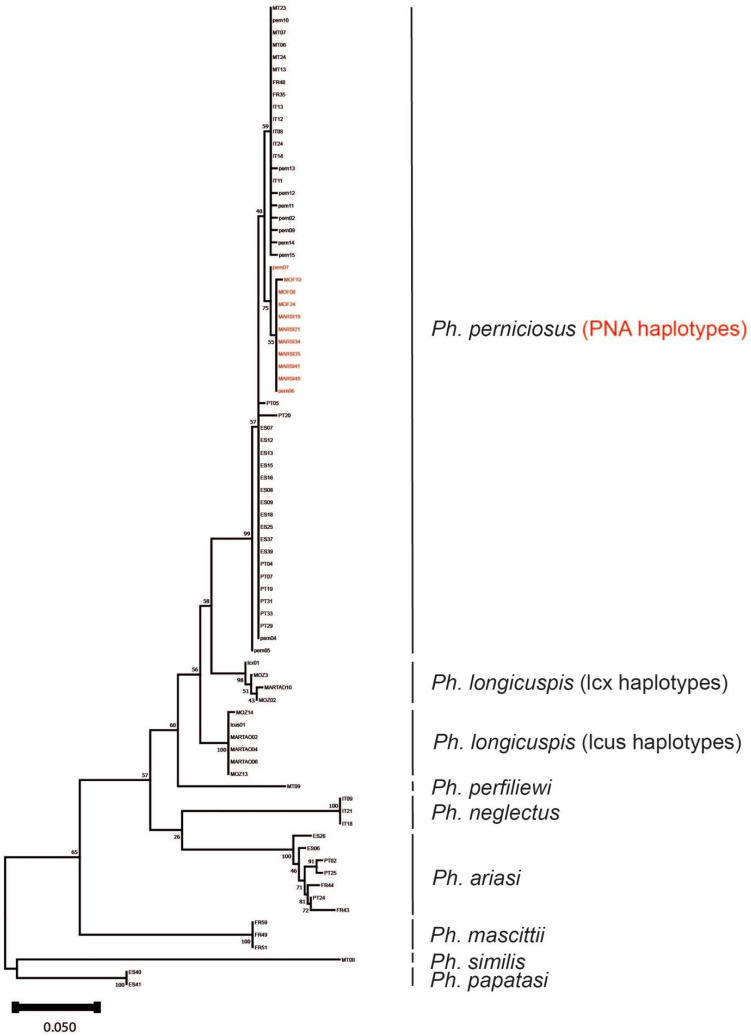
Maximum Likelihood tree based on 84 cyt b sequences using a HKY model [35]. Bootstrap values are indicated on the nodes. Tree was constructed using MEGA11 [30].

**Figure 2 microorganisms-10-02135-f002:**
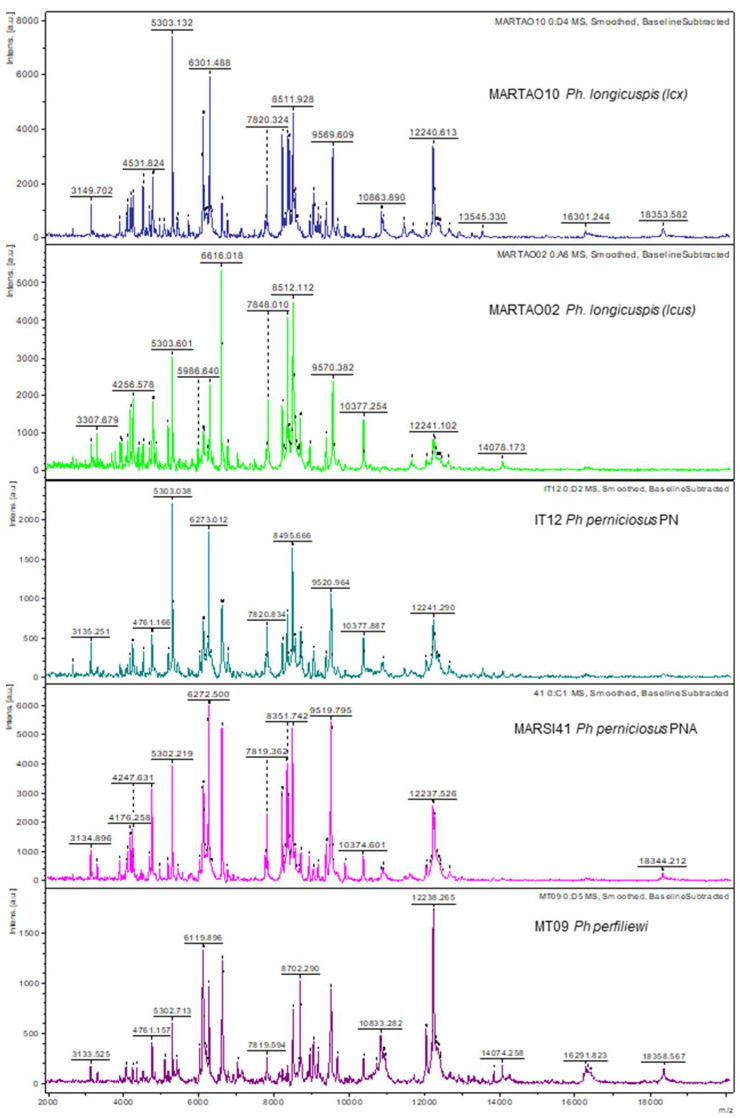
Representative spectra of *Ph. longicuspis* (lcus haplotype), *Ph. longicuspis* (lcx haplotype), *Ph. perniciosus* PN (typical form), *Ph. perniciosus* PNA (atypical form) and *Ph. perfiliewi*.

**Figure 3 microorganisms-10-02135-f003:**
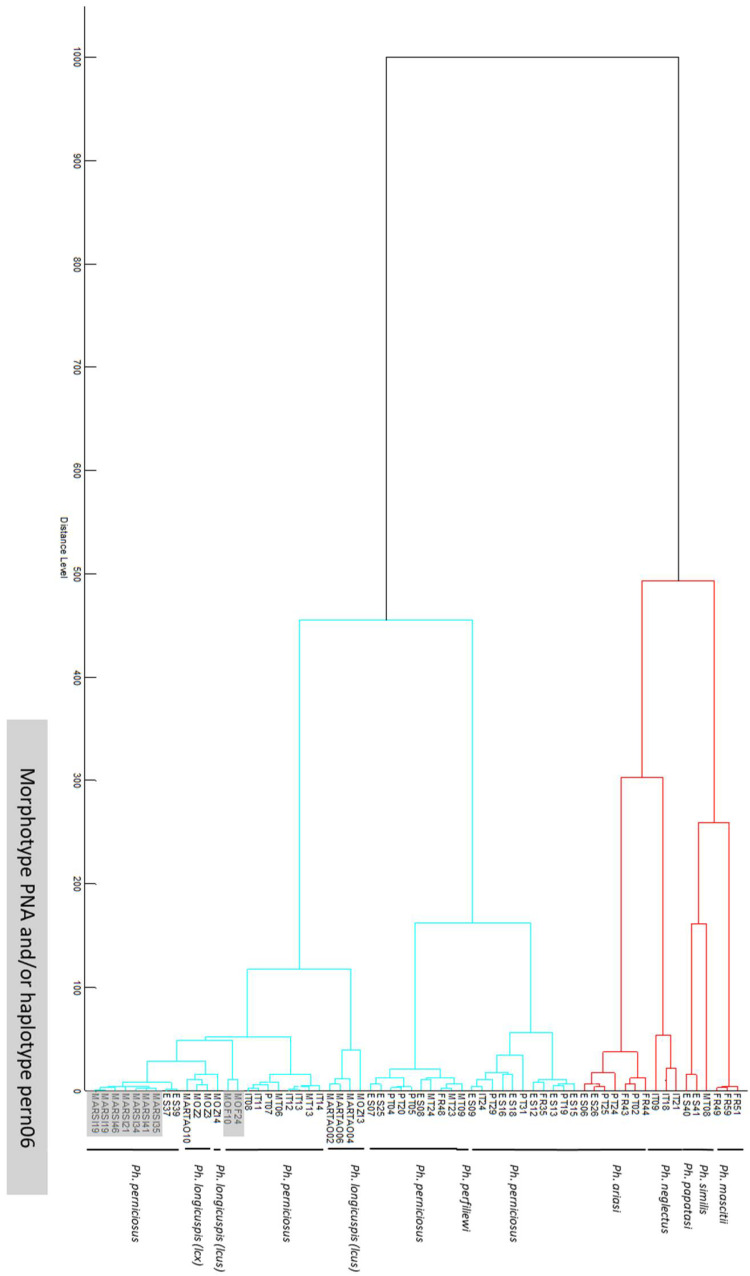
Hierarchical clustering dendrogram of MALDI-TOF MSP using correlation distance measure and Ward algorithm. Molecular identification and PNA haplotype status (grey highlighting) is indicated.

**Figure 4 microorganisms-10-02135-f004:**
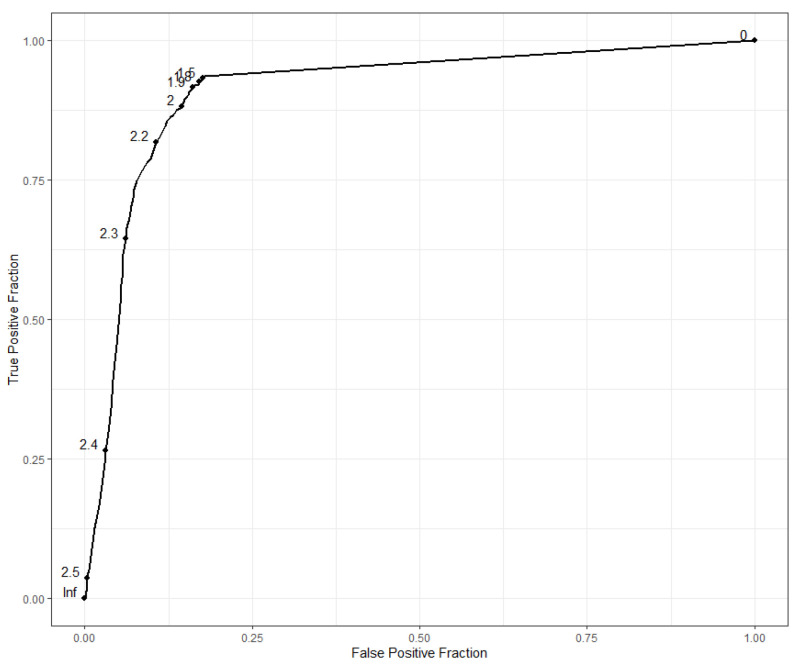
ROC curve analysis of blind test for determination of the optimal LSV cut-off.

**Figure 5 microorganisms-10-02135-f005:**
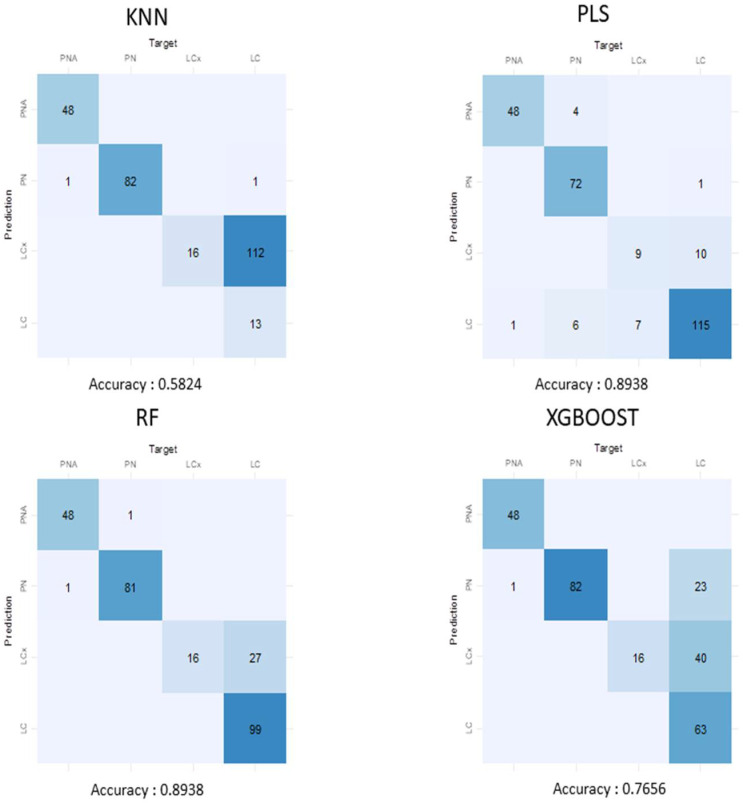
Confusion matrix of various machine learning algorithms for classification of PNA/PN and *Ph. longicuspis* lcx/lcus haplotypes with the test dataset.

**Table 1 microorganisms-10-02135-t001:** Species sampling.

Species	Country	Location	Males	Females
*Ph. perniciosus*	Italy	Gargano (Monte San Angelo)	IT07, IT11, IT14, IT22, IT24	IT08, IT12, IT13,
	France	Touraine (Beauvallon)	FR35	
		Dordogne (Vaunac)	FR47, FR48, FR50	
	Spain	Catalonia (Begues)		ES07
		Saragossa (El Bourgo de Ebro)	ES12, ES13, ES15, ES16	ES08, ES09, ES18, ES18
		Murcia (Verdolay)	ES25, ES29, ES31	ES30
		Andalusia (Turre)	ES37, ES39, ES43, ES45	
	Portugal	Algarve	PT08, PT09, PT13	PT01, PT03, PT04, PT05, PT07, PT19, PT20
		Douro	PT31, PT33, PT34	PT29
	Malta	Gozo	MT01, MT02, MT10, MT11, MT12, MT13, MT21, MT24	MT04, MT05, MT06, MT07, MT23
*Ph. perniciosus* PNA	Morocco	Chefchaouene (Loubar)	MOF8, MOF10	MOF24
		Sidibouyahya	MARSI19, MARSI21, MARSI34, MARSI35, MARSI41, MARSI46	
*Ph. longicuspis*	Morocco	North (Ouezzana Kchile)		MOZ02 (lcx), MOZ03 (lcx), MOZ13 (lcus), MOZ14 (lcus)
		Taounate	MARTAO02 (lcus), MARTAO04 (lcus), MARTAO06 (lcus)	
		Taounate (Aïcha)	MARTAO10 (lcx)	
		Chefchaouene (Loubar)		MOF22 (lcx)
*Ph. neglectus*	Italy	Gargano (Monte San Angelo)	IT09, IT15, IT18, IT21	IT06
*Ph. ariasi*	Spain	Catalonia (Begues)		ES06
		Murcia (Verdolay)		ES26
	Portugal	Algarve		PT02
		Douro		PT24, PT25, PT37
	France	Dordogne (Vaunac)	FR43, FR44	
*Ph. perfiliewi*	Malta	Gozo	MT09	
*Ph. mascittii*	France	Dordogne (Vaunac)		FR49, FR51
		Franche-Comté (Seillières)		FR59
*Ph. papatasi*		Andalusia (Turre)	ES40, ES41	
*Ph. similis*	Malta	Gozo		MT08

**Table 2 microorganisms-10-02135-t002:** Evaluation of the in-house database (LSV cut-off value > 1.7).

	MALDI-TOF MS Identification							
	*Ph. ariasi*	*Ph. mascittii*	*Ph. neglectus*	*Ph papatasi*	*Ph longicuspis (lcus)*	*Ph longicuspis (lcx)*	*Ph. perniciosus*	Total
*Ph. ariasi*	50							50
*Ph. mascittii*		12						12
*Ph. neglectus*			10					10
*Ph papatasi*				8				8
*Ph longicuspis (lcus)*					241	13	1	254
*Ph longicuspis (lcx)*					2	136		138
*Ph. perniciosus*							601	601
Total	50	12	10	8	243	149	601	1073

**Table 3 microorganisms-10-02135-t003:** Results of blind test using Marseille database (LSV cut-off value > 1.7).

	MALDI-TOF MS Identification						
	*Ph. longicuspis*	*Ph. neglectus*	*Ph. papatasi*	*Ph. perfiliewi*	*Ph. perniciosus*	*Ph. tobbi*	Total
*Ph longicuspis (lcus)*	215						215
*Ph longicuspis (lcx)*	137			1			138
*Ph. neglectus*		2					2
*Ph. papatasi*			4				4
*Ph. perniciosus*				13	517	5	535
**Total**	**352**	**2**	**4**	**14**	**517**	**5**	**894**

## Data Availability

Sequences obtained during the present study have been deposited in GenBank under the numbers OP617463 to OP617553, as detailed in Appendix A.

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
