# Peer review of "MALDI-TOF MS Limits for the Identification of Mediterranean Sandflies of the Subgenus Larroussius, with a Special Focus on the Phlebotomus perniciosus Complex"

_microorganisms, 2022, doi:10.3390/microorganisms10112135_

Round 1

Reviewer 1 Report

The objective was to evaluate the performance of MALDI-TOF MS to identify species of the genus Laroussius and to see if this technique can distinguish morphotypes of Ph. perniciosus.

Abstract and introduction is very well described.

In material and methods are samples described, but more information about sundflies is missing. More detailed information about 

Morphological, molecular and mass spectrometry needs more description. More detailed information about conditions is necessary.

Results and discussion is well described, but the conclusion is very general and needs more information about the importance of study. 

Author Response

The objective was to evaluate the performance of MALDI-TOF MS to identify species of the genus Laroussius and to see if this technique can distinguish morphotypes of Ph. perniciosus.

Abstract and introduction is very well described.

In material and methods are samples described, but more information about sundflies is missing. More detailed information about  Morphological, molecular and mass spectrometry needs more description. More detailed information about conditions is necessary.

We have added a phylogenetic tree (Figure 1), permitting the association of different Ph. perniciosus samples to haplotypes. Sequences have been deposited to Genbank. We also add the supplementary table2 which add informations about the specimens used in this study

Results and discussion is well described, but the conclusion is very general and needs more information about the importance of study. 

We add the following paragraph at the end of the conclusion:

The differentiation of these four taxa by morphology requires a trained microscopist and a good dissection of genitalia. The use of chaetotaxy sometimes does not permit to differentiate PNA from Ph longicuspis when the specimen presents 15 or 16 gonocoxal setae. MALDI-TOF is a tool that don’t necessitate formal training in entomology and that can be used to quickly screen a collection of frozen sandflies. However the use of MALDI-TOF in routine testing for closely related sandflies species identification remains difficult at the present time and requires better standardization of the methods used by laboratories worldwide. Morphological expertise remains particularly important and in many cases identification by molecular biology is necessary for reliable diagnosis.

We thank the reviewers for their comments/suggestions which helped to improve this paper.

Reviewer 2 Report

I would like to congratulate with the Authors for this interesting article. It is important to continue research in this field to allow better results in the future. I have appreciated the caution for the interpretation of the results.

Author Response

 Reviewer 2

I would like to congratulate with the Authors for this interesting article. It is important to continue research in this field to allow better results in the future. I have appreciated the caution for the interpretation of the results.

We thank the reviewer 2 and agree that MALDI-TOF is promising but not yet as accurate as molecular biology and results need critical examination. We  highlighted this point in the last sentence of the conclusion:

  Morphological expertise remains particularly important and in many cases identification by molecular biology is necessary for reliable diagnosis.

Reviewer 3 Report

Dear authors,

This is an interesting work about MALDI-TOF use in entomologic applications. However, some questions have to be resolved.

-There are no reasons to include "h" in the acronymous Phlebotomus because there are not any other genera that strart with "P"

-Include numbers of headings and subheadings

-Please include accession numbers of the GenBank

-Include a table with the molecular identification and a phylogenetic tree of the obtained sequences. In the document, there is not any evidence or data about the obtained sequences.

-Improve the quality of figure 2

-The position of P. perfiliewi is mixed with P. perniciosus, but the spectra is very different to the others with several peaks. Maybe the molecular analysis could provide other evidence. If not, which could be the reason? Another reason is that the showed representative spectrum for P. perfiliewi is different from others obtained in the experiment.

-Another recommendation is to use Euclidean distance to construct the dendrogram

Best regards

Author Response

Reviewer 3

This is an interesting work about MALDI-TOF use in entomologic applications. However, some questions have to be resolved.

-There are no reasons to include "h" in the acronymous Phlebotomus because there are not any other genera that strart with "P"

According to Marcondes (2007) then to Galati et al. (2017), the acronymous of sandflies genera are composed of two letters in order to avoid confusion between genera.

Indeed, several genera start with the letter P:

Phlebotomus (Ph.)

Parvidens (Pv.)

Pintomyia (Pi.)

Pressatia (Pr.)

Psathyromyia (Pa.)

Psychodopygus (Ps.)

  1. Marcondes CB. A proposal of generic and subgeneric abbreviations for Phlebotomine sandflies (Diptera: Psychodidae: Phlebotominae) of the World. Entomological News. 2007;118:351-6.

  1. Galati EAB, Galvis-Ovallos F, Lawyer P, Léger N, Depaquit J. An illustrated guide for characters and terminology used in descriptions of Phlebotominae (Diptera, Psychodidae). Parasite. 2017;24:26.

-Include numbers of headings and subheadings

We include the numbers for headings and subheadings, and update the styles for subheadings in material and methods section

-Please include accession numbers of the GenBank

GenBank number have been added

-Include a table with the molecular identification and a phylogenetic tree of the obtained sequences. In the document, there is not any evidence or data about the obtained sequences.

We have added a phylogenetic tree of the obtained sequences with reference sequences from the literature.

-Improve the quality of figure 2

A new high quality version of figure 2 has been added

-The position of P. perfiliewi is mixed with P. perniciosus, but the spectra is very different to the others with several peaks. Maybe the molecular analysis could provide other evidence. If not, which could be the reason? Another reason is that the showed representative spectrum for P. perfiliewi is different from others obtained in the experiment.

Ph. perfiliewi spectra were indeed different in term of peaks intensity however peaks alignement showed a good agreement between the peaks of these two species. As an example, you will attached the peaks alignment of MT09 with the closest MSP from Ph. perfiliewi in the Marseille database (log-score : 2.2150) and with the closest MSP in Reims database (MT09 PN, log-score 2.640). Better alignment is obtained with the PN reference.

-Another recommendation is to use Euclidean distance to construct the dendrogram

We try various dendrogram creation method (correlation, Euclidean, mahanobis distances, cosin etc.) and several linkage algorithm (ward, average etc.). It did not improve the classification compare to correlation with ward algorithm. A dendrogram based on Euclidean distances and ward algorithm is attached.

We are afraid that Ph. perfiliewi and Ph. perniciosus spectra are very close and need more sophisticated techniques such as machine learning to be differentiated.

We thank the reviewers for their comments/suggestions which helped to improve the article.

Round 2

Reviewer 3 Report

Dear authors,

The suggestions have been endorsed correctly. In my opinion, the manuscript can be published in its present form.

Best regards